# Postmortem point-of care hemoglobin testing is feasible and potentially accurate among children in South Africa

Jeanie du Toit[1], Yuqing Wang[2], Hanqi Luo[2], Lei Liu[2], Dianna M. Blau[3], Cynthia G. Whitney[2], Rochelle Werner[2], Quique Bassat[4,5,6,7], Kimberleigh Storath[1], Palesa Makekeng[1], Ziyaad Dangor[1], Shabir A. Mahdi[1,8], Valentine Wanga[2], Parminder S. Suchdev[2,3,9]*

1 South African Medical Research Council, Vaccines and Infectious Diseases Analytics Research Unit, University of the Witwatersrand, Johannesburg, South Africa, 2 Hubert Department of Global Health, Emory University, Atlanta, Georgia, United States of America, 3 Global Health Center, US Centers for Disease Control and Prevention, Atlanta, Georgia, United States of America, 4 ISGlobal, Hospital Clínic - Universitat de Barcelona, Barcelona, Spain, 5 Centro de Investigação em Saúde de Manhiça (CISM), Maputo, Mozambique, 6 ICREA, Barcelona, Spain, 7 CIBER de Epidemiología y Salud Pública, Instituto de Salud Carlos III, Madrid, Spain, 8 Wits Infectious Diseases and Oncology Research Institute, Faculty of Health Sciences, University of the Witwatersrand, Johannesburg, South Africa, 9 Department of Pediatrics, Emory University, Atlanta, Georgia, United States of America

* psuchde@emory.edu

## Abstract

Anemia is an important cause of child morbidity and mortality. Postmortem point-of-care hemoglobin testing is a potential method for assessing anemia at death, but its reliability has not been extensively studied. We aimed to assess the feasibility and validity of postmortem point-of-care hemoglobin assessment using HemoCue in the setting of a child mortality surveillance program in South Africa. In a pilot cohort study, 44 children under five years of age who died in an academic hospital in South Africa were enrolled. Hemoglobin levels were measured from venous blood antemortem using standard hematology analyzers and postmortem using the HemoCue 201 from blood collected within 72 hours of death (either by needle aspiration or from whole blood collected in an EDTA tube). Updated World Health Organization hemoglobin cutoffs to define anemia were used. Wilcoxon signed-rank tests, equivalence tests, and regression models assessed the concordance between antemortem and postmortem hemoglobin concentrations. Postmortem testing showed a significant decrease in hemoglobin concentrations compared to antemortem levels. However, no significant differences were found between hemoglobin measurements from needle aspiration and those from EDTA tubes postmortem. The prevalence of anemia increased from 52% antemortem to 73–77% postmortem, with the most notable rises in moderate and severe anemia. Bland-Altman analysis confirmed a systematic, not random, decrease in postmortem hemoglobin measurements. Upon applying a fixed adjustment of 2.5 g/dL, the sensitivity and specificity of postmortem hemoglobin testing to diagnose anemia were 69.6% and 61.9%, respectively. Postmortem point-of-care hemoglobin testing using HemoCue is feasible and offers a potentially valid reflection of antemortem anemia status in deceased children, despite consistently lower measured

**Data availability statement:** All relevant data are within the paper and its Supporting Information files. The original data is from: https://champshealth.org/data/

**Funding:** This work was funded by the Bill & Melinda Gates Foundation in the form of grants [OPP1126780 to CGW; INV049213 to PSS]. The funders had no role in study design, data collection and analysis, decision to publish, or preparation of the manuscript.

**Competing interests:** The authors have declared that no competing interests exist.

values postmortem. These findings support the utility of postmortem hemoglobin assessments in determining the presence and severity of anemia at the time of death.

## Introduction

The reduction in childhood mortality remains a public health priority. Although there has been a significant decrease in deaths of children under five years old, the global under-5 mortality rate still exceeds the Sustainable Development Goal target of 25 deaths per 1000 livebirths (SDG 3.2) [1]. Anemia affects one-third of the world's population and is not only associated with impaired cognitive and behavioral development in children, but severe anemia is recognized as a direct and important, albeit possibly under recognized, cause of child mortality [2–4]. Anemia, characterized as a reduced absolute number of circulating red blood cells or a condition in which the red blood cells present cannot adequately meet physiological oxygen demands, is defined by low hemoglobin concentrations [5]. Sub-Saharan Africa has the highest prevalence of anemia compared with any other region, primarily due to the disproportionate infection burden and the low bioavailability of dietary iron [6]. Anemia itself is not a disease but a condition indicative of diverse and overlapping etiologies, including nutritional deficiencies, genetic factors, infections, and environmental influences [7]. Thus, understanding the underlying causes of anemia is crucial for its prevention and treatment in both clinical and public health contexts.

Children under five, particularly infants and toddlers under two years of age, are the most susceptible to anemia [8]. The estimated proportion of deaths due to anemia from primarily hospital-based studies in malarial areas ranges from 3% to 58%; fatality is increased for those with severe anemia [9]. The Child Health and Mortality Surveillance (CHAMPS) study offers the advantage of attributing underlying, immediate, and antecedent causes of death [10], achieving a level of granularity in identifying these causes that is seldom available through other surveillance efforts. Gaining a broader understanding of the causes of mortality in children under 5 and how anemia contributes to these deaths is crucial for identifying effective interventions.

Automated hematology analyzers or spectrometry have been used reliably to measure hemoglobin from venous and capillary samples in both clinical and public health settings [7]. HemoCue is a portable device based on spectrometry that has acceptable performance when compared with automated hematology analyzers and is thus routinely used for rapid, point-of-care hemoglobin assessment [11]. Given emerging evidence that suggests the use of single-drop capillary samples can introduce random and systematic error and lead to inaccurate estimates of anemia, new global guidelines suggest using venous or pooled capillary blood when assessing hemoglobin concentration in individuals and populations [12–14].

Measuring hemoglobin postmortem for the diagnosis of anemia may present several challenges such as postmortem changes in hemoglobin concentrations due to hemolysis, declines in circulating blood volume due to hemorrhage or hypovolemic shock at the time of death, and loss of integrity of blood samples due to prolonged time since death [15]. While a prior study suggested that point-of-care hemoglobin testing may be reliable for postmortem diagnosis of anemia [16], to our knowledge, no study has assessed the utility and reliability of postmortem hemoglobin testing using HemoCue. CHAMPS provides a unique platform to assess hemoglobin measurements in children during the antemortem and postmortem periods. Comparisons of antemortem and postmortem hemoglobin measurements could help determine whether anemia was present at the time of death for deaths that occurred outside healthcare settings or before hemoglobin could be measured in a health facility; such information could inform interventions designed to reduce the burden of anemia in children.

Thus, this study aimed to assess the feasibility and validity of postmortem point-of-care hemoglobin assessment using HemoCue in the setting of the CHAMPS platform in South Africa. Further, the study assessed whether pre-analytical factors such as timing of blood collection and measurement and sources of postmortem venous blood were associated with differences in measured hemoglobin concentration.

## Materials and methods

### Study design, setting, and participants

A pilot cohort study was conducted at Chris Hani Baragwanath Academic Hospital, South Africa, between July 2023 and June 2024. Children were included in the study, and postmortem hemoglobin testing was conducted if the following criteria were met: 1) enrolled in the CHAMPS study, were liveborn and demised before five years of age; 2) had results of hemoglobin measures conducted antemortem; 3) postmortem samples were collected within 72 hours of death. Details of the CHAMPS methodology have been described previously [17]. In brief, a consenting team was dispatched to the family following a child's death to confirm eligibility and obtain written consent for all CHAMPS procedures. In this pilot study, to calculate the necessary sample size, we assumed hemoglobin values would decrease antemortem to postmortem. Assuming mean antemortem hemoglobin of 10.0 g/dL, a sample size of 25 would produce 80% power to detect a 0.8 g/dL difference between antemortem and postmortem measurements. Ethical approval was obtained from institutional review boards of University of the Witwatersrand and Emory University Rollins School of Public Health. A total of 44 participants were enrolled in our study.

### Lab methods

Antemortem testing of hemoglobin was performed using venous blood and a standard hematology analyzer (Sysmex XN-10 or Sysmex XN-20, Japan). Antemortem samples were collected as part of routine blood work performed on children during their admission and hospital course. The number of samples drawn varied and were based on the treating clinician's discretion. For children with multiple antemortem hemoglobin values, the final 3 were recorded, and the measurement closest to time of death was used for analysis.

Postmortem blood samples were collected during the minimally invasive tissue sampling (MITS) procedure following death with blood drawn from an intra-cardiac or subclavian artery puncture less than 72 hours after death from the refrigerated body. This process has been thoroughly described previously [18]. Anonymity was strictly maintained, the identities of the participants were never disclosed, and the information gathered remained confidential. Following needle aspiration, whole blood was tested immediately using the HemoCue 201+ (Angelholm, Sweden) using a single drop of blood on wax paper. The rest of the sample was then transferred into an EDTA tube. Approximately 1 hour after collection of the initial sample, the EDTA blood sample was tested using the HemoCue 201+ by tilting and directly placing the cuvette into the EDTA tube. The dates and times of blood collection and testing, time of death, any antemortem blood transfusions, and the appearance of sampled blood (e.g., normal, hemolyzed, clotted) were recorded.

### Statistical methods

Data were analyzed using R version 4.2.1 (2022-06-23). The severity of anemia was defined based on unadjusted hemoglobin concentration as follows: no anemia (hemoglobin ≥ 10.5 g/dL for children < 24 months and ≥ 11.0 g/dL for children ≥ 24 months), mild anemia (hemoglobin between 9.5 - 10.4 g/dL for children < 24 months and between 10.0 - 10.9 g/dL for

children ≥ 24 months), moderate anemia (hemoglobin between 7.0 - 9.4 for children < 24 months and between 7.0 - 9.9 g/dL for children ≥ 24 months), and severe anemia (hemoglobin <7.0 g/dL) [13]. Mean and standard deviation for continuous variables and percentage for categorical variables were calculated. A Wilcoxon signed-rank test was used to assess differences in timing between hemoglobin tests and death and in hemoglobin concentrations between antemortem and postmortem tests. Additionally, the McNemar test was applied to detect differences between any anemia and no anemia, and the Stuart-Maxwell test was applied to detect differences in the rest of the anemia categories between antemortem and postmortem tests. The significance level was set at $p < 0.05$.

Box plots were used to illustrate the distribution of anemia severity across various hemoglobin tests. An Alluvial plot was presented to show changes in hemoglobin concentrations from the last antemortem test to postmortem testing via needle aspiration and EDTA. Additionally, a Bland-Altman plot was presented to assess the agreement between antemortem tests and postmortem measurements using EDTA. To further investigate the use of postmortem EDTA hemoglobin to diagnose anemia, the sensitivity and specificity was calculated by comparing it against antemortem hemoglobin concentration measured by a hematology analyzer.

## Results

A total of 44 children were included in the analysis; their mean age was 4.5 months (range: 0 days – 47.1 months, Table 1). The average time between antemortem hemoglobin testing and death was 36.5 hours (range: 0.3 – 137.7 hours), and the average time between death and postmortem testing with needle aspiration was 22.5 hours (range: 5.5 – 51.9 hours). Hemoglobin testing with the EDTA tube was performed on average within 1 hour (range: 0.3 – 1.2 hours) of the testing with needle aspiration. The quality of postmortem blood samples was good with only approximately 11% clotted or diluted, as compared with 32% of antemortem samples showing signs of clotting (S2 Table). There was a statistically significant reduction in hemoglobin concentrations when comparing antemortem and postmortem values ($p < 0.001$ between antemortem vs needle aspiration and antemortem vs EDTA); however, there was

**Table 1. Distribution of age, testing interval, hemoglobin, and anemia severity\* by hemoglobin assessment method among children under five years, CHAMPS South Africa from July 2023 to June 2024 (n = 44).**

| Variable | Antemortem | Postmortem | | P-value | | |
|---|---|---|---|---|---|---|
| | | Needle Aspiration | EDTA Tube | Antemortem & Needle Aspiration | Antemortem & EDTA | Needle Aspiration & EDTA |
| Age (mo), mean (SD) | 4.5 (10.8) | | | | | |
| Time between death and testing (hours), mean (SD) | 36.5 (38.9) | 22.5 (11.5) | 23.0 (11.4) | | | |
| Hemoglobin (g/dL), mean (SD) | 10.8 (3.4) | 7.9 (4.1) | 8.3 (4.3) | <0.001 | < 0.001 | 0.29 |
| Any anemia, % (n) | 52.3 (23) | 76.7 (33) | 72.7 (32) | 0.034 | 0.027 | 0.99 |
| No anemia, % (n) | 47.7 (21) | 23.3 (10) | 27.3 (12) | | | |
| Mild anemia, % (n) | 18.2 (8) | 4.5 (2) | 2.3 (1) | 0.005 | 0.001 | 0.91 |
| Moderate anemia, % (n) | 22.7 (10) | 31.8 (14) | 29.5 (13) | | | |
| Severe anemia, % (n) | 11.4 (5) | 39.5 (17) | 40.9 (18) | | | |

\* Anemia defined based unadjusted hemoglobin concentration: no anemia (≥ 10.5 g/dL for children < 24 months and ≥ 11.0 g/dL for children ≥ 24 months), overall (< 10.5 g/dL for children < 24 months and < 11.0 g/dL for children ≥ 24 months), mild (9.5-10.4 g/dL for children < 24 months and 10.0-10.9 g/dL for children ≥ 24 months), moderate (7.0-9.4 g/dL for children < 24 months and 7.0-9.9 g/dL for children ≥ 24 months), and severe (<7.0 g/dL). Wilcoxon signed-ranks test was used to detect time differences between testing and death and hemoglobin concentrations. The McNemar test was used for any anemia category, and Stuart-Maxwell test was used for the rest of anemia categories for Antemortem vs Needle, and Antemortem vs EDTA, and Needle vs EDTA.

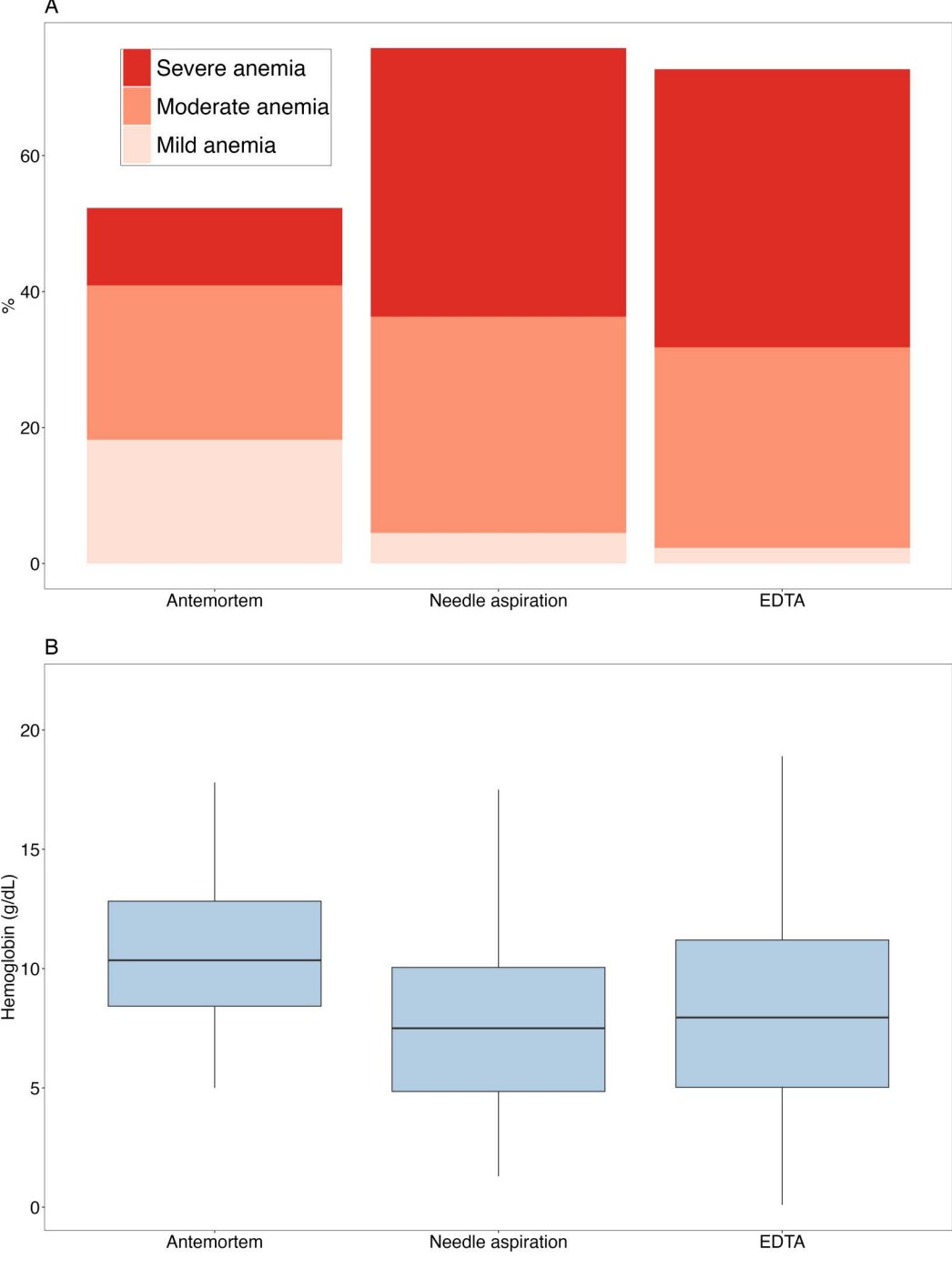

**Fig 1. A) Proportion of subjects with hemoglobin measurements meeting definitions for severe, moderate, and mild anemia based on unadjusted measurements", B) Box-Whisker plot of hemoglobin distribution using different methods (antemortem, postmortem needle aspiration, and EDTA tube).**

no difference between postmortem needle aspiration and EDTA hemoglobin measurements (Table 1, Fig 1). The prevalence of any anemia increased postmortem compared with antemortem (73% EDTA and 77% needle aspiration postmortem vs. 52% antemortem). When

stratified by anemia severity, the prevalence of moderate, and severe anemia increased postmortem when comparing antemortem values to both needle aspiration and EDTA-derived postmortem values (Fig 1).

Fig 2 is an alluvial plot that shows changes in individual hemoglobin concentrations over time. For most cases (75%), the postmortem hemoglobin concentration was lower than the antemortem hemoglobin concentration, irrespective of whether the child received a blood transfusion and regardless of the time between antemortem testing and death. A linear regression model confirmed no association between changes in antemortem and postmortem hemoglobin concentrations and the time elapsed between death and postmortem testing nor the time between antemortem and postmortem testing (p = 0.92 and p = 0.34, respectively, S2 Table).

A Bland-Altman analysis was used to assess the agreement between antemortem and postmortem EDTA measurements (Fig 3). Postmortem hemoglobin concentrations were, on average, 2.5 g/dL lower than those obtained antemortem. Further, nearly all data points, including those cases who received a blood transfusion, fell within two standard deviations, suggesting systematic error and not random error.

Given the apparent fixed difference between antemortem and postmortem hemoglobin values, we next calculated the sensitivity and specificity for postmortem hemoglobin testing to diagnose anemia using unadjusted and adjusted postmortem hemoglobin values (Fig 4). Using unadjusted values, the sensitivity and specificity of postmortem hemoglobin concentrations to detect any level of anemia were 91.3% and 47.6%, respectively. After applying a fixed adjustment of 2.5 g/dL, the sensitivity to diagnosis anemia decreased to 69.6%, but the specificity increased to 61.9%. For severe anemia, there was a similar pattern in which sensitivity decreased (60% to 0%), but specificity increased (61.5% to 79.5%); making the adjustment,

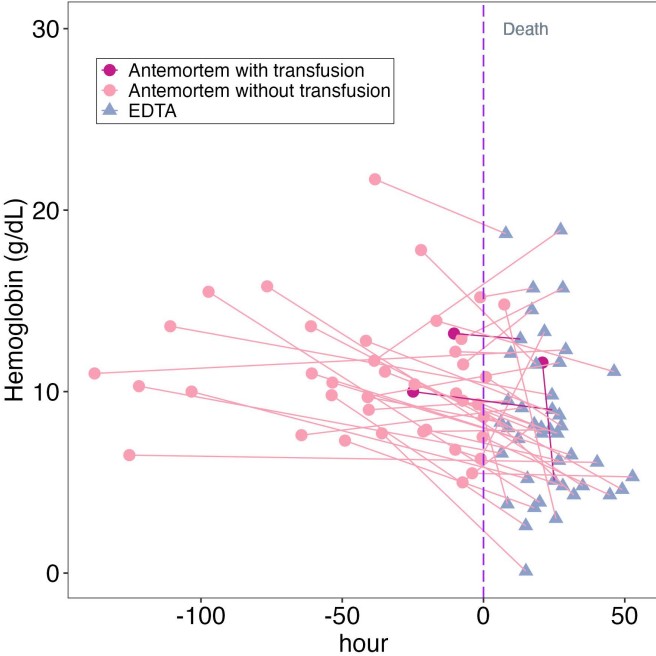

**Fig 2. Alluvial plot of hemoglobin concentrations antemortem and postmortem using ETDA testing by time before or after death.**

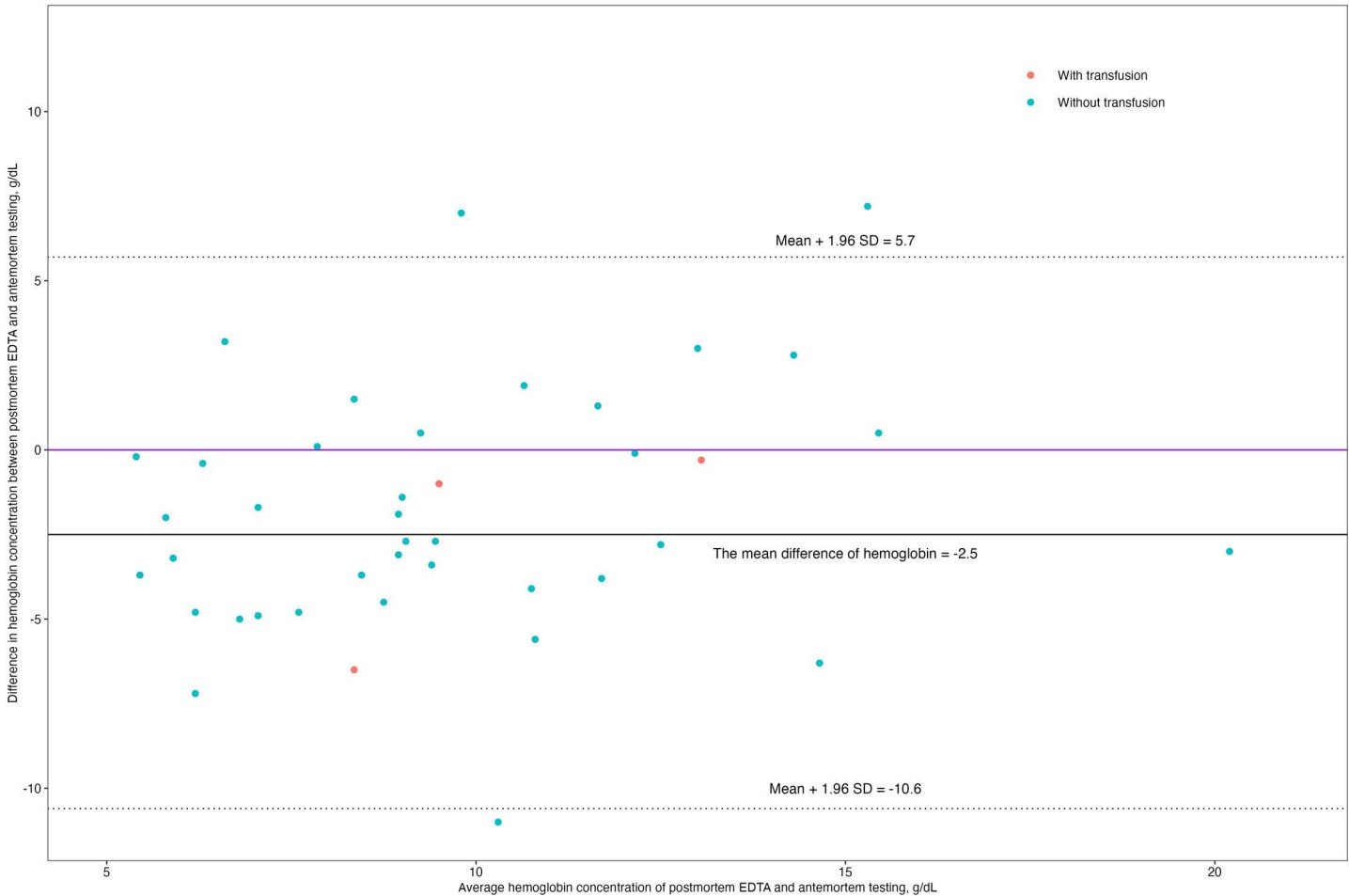

**Fig 3. Bland-Altman plot of hemoglobin concentrations measured via antemortem and postmortem EDTA method.**

however, resulted in none of the 5 deaths with severe anemia based on antemortem measurements classified as severe anemia based on postmortem measurements.

## Discussion

This pilot evaluation of postmortem point-of-care hemoglobin testing using HemoCue represents a feasible and potentially useful method for assessing antemortem anemia status in deceased children and evaluating the potential role of anemia in under-5 deaths. Although there was a decrease in postmortem hemoglobin measurements compared to antemortem values, the error appeared to be systematic rather than random. While anemia is an important and often preventable cause of death [19], it is often under-reported as a contributor to the cause of death, particularly in low- and middle-income countries (LMICs) where antemortem testing is not accessible, and many deaths occur outside of hospital settings. Accurately assessing hemoglobin status in under-5 deaths may also help identify individuals and populations who could benefit from targeted interventions based on demographics, genetic, or environmental factors [7]. Point-of-care hemoglobin testing has previously proven to be useful for diagnosis of anemia and has been advocated for use as an adjunct to full postmortem examinations when deemed necessary by the pathologists [16].

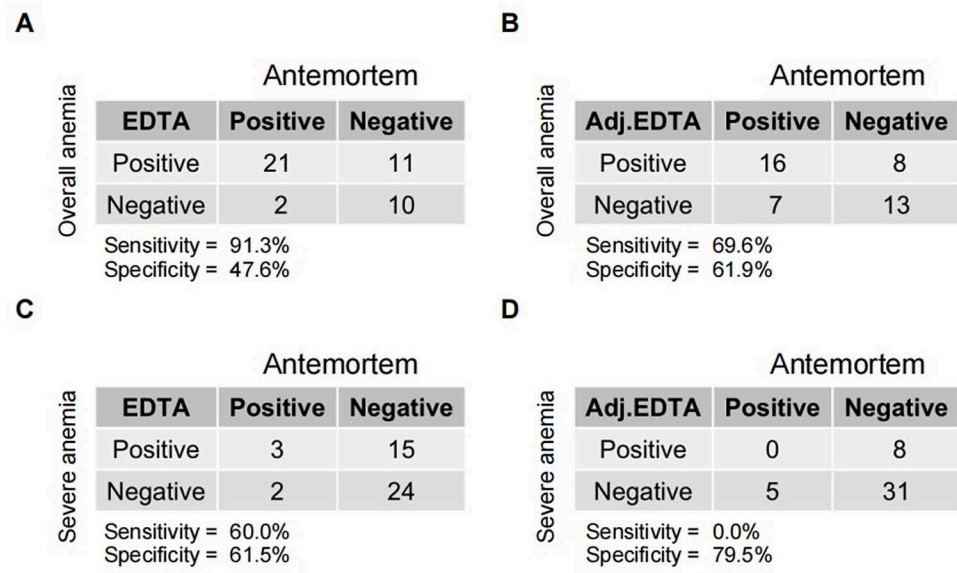

**Fig 4. Sensitivity and specificity of using postmortem hemoglobin concentrations to diagnosis anemia.**

The feasibility of a diagnostic approach is critically important, particularly in LMICs. This study employed a well-described and straightforward protocol, ensuring that all procedures were effectively conducted and can be easily replicated. Portable point-of-care devices like the HemoCue are generally affordable and easy to use in both clinical laboratory and field settings [20]. Staff can be easily trained to use a HemoCue without medical expertise. In this study, both needle aspiration and EDTA hemoglobin measurements yielded equivalent concentrations. This equivalence is crucial for the feasibility of the study, as either method can be adopted depending on the setting. Both approaches were equally accepted by the study staff, supporting the viability of performing measurements through either approach.

The Bland-Altman analysis from this study confirmed the accuracy of the HemoCue device in measuring hemoglobin in the postmortem setting, showing modest agreement with antemortem values. Previous studies have highlighted that both systematic and random error can affect hemoglobin concentration measurements using HemoCue [21]. In this study, the systematic error, which is constant and impacts all results equally, was mitigated by mathematically adjusting the postmortem hemoglobin values—specifically by adding a fixed 2.5 g/dL—to better align them with reference antemortem values obtained from gold standard automated hematology analyzers [21]. The observed decrease in postmortem hemoglobin compared to antemortem hemoglobin was sizeable but relatively consistent, likely reflecting the progression of the disease leading to death (e.g., sepsis or hemolysis), time-dependent postmortem changes in blood composition, or systematic differences between the autoanalyzer and HemoCue rather than random artifact or measurement error. Depending on the severity and trajectory of the disease, hemoglobin concentrations may have been declining before death, thus limiting the correlation between antemortem and postmortem hemoglobin levels. The sensitivity and specificity of postmortem hemoglobin testing for diagnosing anemia were modest, suggesting that while hemoglobin testing alone may not be sufficient for clinical diagnosis, it can still provide useful information for clinicians assessing the contribution of anemia to child mortality. Importantly, the presence of anemia does not necessarily imply causation in child mortality. Anemia is a condition reflective of diverse and overlapping causes and

**Table 2. Comparison of antemortem and unadjusted/adjusted postmortem hemoglobin concentrations\* among deaths in children under five years of age, CHAMPS South Africa from July 2023 to June 2024 (n = 44).**

| | Antemortem Testing | Postmortem Needle Aspiration | | | Postmortem EDTA | | |
|---|---|---|---|---|---|---|---|
| | | Unadjusted | Adjusted | P-value[+] | Unadjusted | Adjusted | P-value[+] |
| Hemoglobin (g/dL), mean (SD) | 10.8 (3.4) | 7.9 (4.1) | 10.4 (4.1) | 0.64 | 8.3 (4.3) | 10.8 (4.3) | 0.9 |
| Any anemia, % (n) | 52.3 (23) | 76.7 (33) | 55.8 (24) | 1.0 | 72.7 (32)) | 50.0 (22) | 1.00 |
| No anemia, % (n) | 47.7 (18) | 23.3 (10) | 44.2 (19) | | 27.3 (12) | 50.0 (22) | |
| Mild anemia, % (n) | 18.2 (8) | 4.5 (2) | 15.9 (7) | 0.6 | 2.3 (1) | 9.1 (4) | 0.54 |
| Moderate anemia, % (n) | 22.7 (10) | 31.8 (14) | 15.9 (7) | | 29.5 (13) | 22.7 (10) | |
| Severe anemia, % (n) | 11.4 (5) | 39.5 (17) | 23.3 (10) | | 40.9 (18) | 18.2 (8) | |

\* Anemia defined based unadjusted/adjusted hemoglobin concentration: no anemia (≥ 10.5 g/dL for children < 24 months and ≥ 11.0 g/dL for children ≥ 24 months), overall (< 10.5 g/dL for children < 24 months and < 11.0 g/dL for children ≥ 24 months), mild (9.5-10.4 g/dL for children < 24 months and 10.0-10.9 g/dL for children >= 24 months), moderate (7.0-9.4 g/dL for children < 24 months and 7.0-9.9 g/dL for children≥ 24 months), and severe (<7.0 g/dL). Hemoglobin concentrations were adjusted by adding the mean difference between antemortem and EDTA hemoglobin concentrations (2.5 g/dL) to the original, unadjusted values. The McNemar test was used for any anemia category, and Stuart-Maxwell test was used for the rest of anemia categories for antemortem vs adjusted needle and antemortem vs adjusted EDTA.

+ P values indicated the statistical significance of differences between antemortem and adjusted postmortem hemoglobin concentrations.

underlying pathological processes, such as severe infections, nutritional deficiencies, or chronic illnesses, which ultimately drive the progression to death.[7,19] Thus, distinguishing between anemia as a primary cause and as a secondary consequence of other conditions is critical for accurate interpretation. While postmortem hemoglobin testing provides valuable data, it must be contextualized within a comprehensive assessment of the child's clinical and pathological history to ascertain the causal pathways leading to death. Findings from this pilot study have prompted the expansion of postmortem hemoglobin testing using HemoCue at other CHAMPS surveillance sites in South Asia and Sub-Saharan Africa to better quantify and assess the role anemia on child mortality.

As this was a pilot study, one significant limitation is the small sample size, gathered from a single community in Soweto, South Africa. With increased sample size, it will be important to evaluate potential non-linear associations between antemortem and postmortem hemoglobin concentrations, which may require a variable rather than a fixed correction factor. Second, while previous studies have evaluated the comparability of HemoCue and automated hematology analyzers, we are unable to assess potential systematic differences between these techniques since we did not measure both antemortem and postmortem hemoglobin concentrations using both an autoanalyzer and HemoCue. Third, it is important to note that the study team prioritized specificity over sensitivity, especially for severe anemia. As a result, all five cases with severe anemia based on antemortem hemoglobin concentrations were no longer diagnosed with severe anemia using adjusted postmortem values. Despite these limitations, the study has notable strengths, including the comparison of postmortem samples with antemortem samples measured via venous blood and the gold standard automated hematology analyzer. Additionally, the time taken to test the postmortem samples was relatively short, as the CHAMPS protocol requires the MITS procedure to be completed within 72 hours of death, often within 24 hours.

## Conclusions

This pilot study was the first study, to our knowledge, to assess hemoglobin measurements postmortem using the point-of-care HemoCue device. The results are encouraging, suggesting that this method may be useful for estimating antemortem hemoglobin concentrations and

evaluating the role of anemia as a contributing factor in the causal pathway of death. These findings contribute to global health by demonstrating the value of postmortem hemoglobin testing as a practical tool to improve cause-of-death reporting in settings with limited access to healthcare and diagnostics, thereby informing future interventions for the prevention and treatment of anemia. Future studies involving larger numbers of deaths could help determine the appropriate correction factor for the systematic error observed and provide insight into the mechanisms behind postmortem decreases in hemoglobin concentrations. In addition, further research is needed to evaluate the nutritional and non-nutritional causes of anemia to guide more effective interventions.

## Supporting information

**S1 Table. Association between the changes in hemoglobin concentrations measured via postmortem EDTA method and time** .
(DOCX)

**S2 Table. Summary statistics of blood status by different testing methods among children under five years, CHAMPS South Africa, July 2023 to June 2024 (n = 44).**
(DOCX)

**S1 Data. CHAMPS pilot Hb dataset.**
(CSV)

## Author contributions

**Conceptualization:** Hanqi Luo, Valentine Wanga, Parminder S. Suchdev.

**Data curation:** Jeanie du Toit, Dianna M Blau.

**Formal analysis:** Yuqing Wang, Hanqi Luo, Lei Liu.

**Investigation:** Jeanie du Toit, Kimberleigh Storath, Palesa Makekeng.

**Methodology:** Yuqing Wang, Hanqi Luo.

**Project administration:** Jeanie du Toit, Dianna M Blau, Cynthia G. Whitney, Rochelle Werner, Valentine Wanga.

**Resources:** Cynthia G. Whitney.

**Supervision:** Hanqi Luo, Ziyaad Dangor, Shabir A Mahdi, Parminder S. Suchdev.

**Visualization:** Yuqing Wang.

**Writing – original draft:** Jeanie du Toit, Yuqing Wang, Parminder S. Suchdev.

**Writing – review & editing:** Hanqi Luo, Dianna M Blau, Cynthia G. Whitney, Rochelle Werner, Quique Bassat, Kimberleigh Storath, Palesa Makekeng, Ziyaad Dangor, Valentine Wanga, Shabir A Mahdi, Parminder S. Suchdev.

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
