## [Decision Letter · Decision Letter 0]

24 Oct 2024

PGPH-D-24-01978

Postmortem point-of care hemoglobin testing is feasible and potentially accurate among children in South Africa

Dear Dr. Parminder Suchdev,

Thank you for submitting your manuscript to PLOS Global Public Health. After careful consideration, we feel that it has merit but does not fully meet PLOS Global Public Health’s publication criteria as it currently stands. Therefore, we invite you to submit a revised version of the manuscript that addresses the points raised during the review process.

The three referees agree that the manuscript provides a promising analysis and potentially of interest, but they also identify several issues. They have provided specific recommendations to that end. Additionally, the data utilized in this study are not openly available to public. Please note that one of our publication criteria is the following. 7. The article adheres to appropriate reporting guidelines and community standards for data availability.  I would be glad to consider a revised version of your manuscript that takes these comments and suggestions into account.

We look forward to receiving your revised manuscript.

Kind regards,

Masako Fujita, Ph.D.

Academic Editor

Journal Requirements:

1. Please provide additional details regarding participant consent. In the ethics statement in the Methods and online submission information, please ensure that you have specified (1) whether consent was informed and (2) what type you obtained (for instance, written or verbal, and if verbal, how it was documented and witnessed). If your study included minors, state whether you obtained consent from parents or guardians. If the need for consent was waived by the ethics committee, please include this information.

3. Please include a complete copy of PLOS’ questionnaire on inclusivity in global research in your revised manuscript. Our policy for research in this area aims to improve transparency in the reporting of research performed outside of researchers’ own country or community. The policy applies to researchers who have travelled to a different country to conduct research, research with Indigenous populations or their lands, and research on cultural artefacts. The questionnaire can also be requested at the journal’s discretion for any other submissions, even if these conditions are not met.  Please find more information on the policy and a link to download a blank copy of the questionnaire here: https://journals.plos.org/globalpublichealth/s/best-practices-in-research-reporting. Please upload a completed version of your questionnaire as Supporting Information when you resubmit your manuscript.

4. Please provide separate figure files in .tif or .eps format.

5. We do not publish any copyright or trademark symbols that usually accompany proprietary names, eg (R), (C), or TM  (e.g. next to drug or reagent names). Please remove all instances of trademark/copyright symbols throughout the text, including © on page 22 and ® on pages 8 and 23.

Reviewers' comments:

Reviewer's Responses to Questions

**Comments to the Author**

1. Does this manuscript meet PLOS Global Public Health’s publication criteria ? Is the manuscript technically sound, and do the data support the conclusions? The manuscript must describe methodologically and ethically rigorous research with conclusions that are appropriately drawn based on the data presented.

Reviewer #1: Yes

Reviewer #2: Yes

Reviewer #3: Yes

2. Has the statistical analysis been performed appropriately and rigorously?

Reviewer #1: Yes

Reviewer #2: Yes

Reviewer #3: Yes

3. Have the authors made all data underlying the findings in their manuscript fully available (please refer to the Data Availability Statement at the start of the manuscript PDF file)?

Reviewer #1: Yes

Reviewer #2: No

Reviewer #3: No

4. Is the manuscript presented in an intelligible fashion and written in standard English?

Reviewer #1: Yes

Reviewer #2: Yes

Reviewer #3: Yes

5. Review Comments to the Author

Reviewer #1: This article, “Postmortem point-of care hemoglobin testing is feasible and potentially accurate

among children in South Africa,” describes the validation of point-of-care HemoCue testing for hemoglobin in deceased infants and children. They compared HemoCue samples with whole blood sampling and antemortem hemoglobin samples and determined that values are systematically lower post-death. They conclude that this method can be used for determining anemia postmortem.

I do not have much issue with the methods or analysis. My suggestion for revision is to spend more space discussing why this measure is potentially useful for global health, and to write in more depth about the causes of variation in hemoglobin levels postmortem and how hemoglobin changes in severe (death-causing) morbidity. It’s not clear to me whether this method hopes to find anemia caused by malnutrition or anemia that results from another illness that hospitalized the infant. Any control for the cause of death? How close to time of death were the antemortem samples collected?

Reviewer #2: Overall, this is a well-written research article that provides compelling data on the utility of postmortem point-of-care hemoglobin testing using HemoCue. The authors present the importance of understanding a child’s anemia status to better understand the potential factors contributing to their death. The research methodology is straightforward and the results suggest that applicability of the proposed method for the postmortem evaluation of anemia status. The only substantive comments that I have are regarding the discussion section. I think it would be valuable to provide a short discussion about whether any error between the antemortem and postmortem hemoglobin values could be caused by differences in diagnostic technique (i.e. standard hematology analyzers and HemoCue). I also am curious as to the authors’ interpretation of the 22% of cases where the postmortem hemoglobin concentrations were higher than the antemortem concentrations. Would it also be important to test the association between changes in antemortem and postmortem concentrations and the time elapsed between antemortem and postmortem testing (rather than only the time between death and postmortem testing)? Also in the last paragraph of the Discussion, third sentence, it states, “all three cases with severe anemia,” however in the results and Table 1 there are five cases with severe anemia. Please see attached document for grammatical revisions.

Reviewer #3: These analyses are sound and the conclusions justified. I recommend the authors include a discussion of the broader implications of their work, and how the method they propose (using HemoCue devices to identify anemia postmortem) can be used wisely, including the risk for misattribution of cause and effect (see notes for more details). I also think they should flag for reviewers places where their analyses might extend beyond what was contemplated in their original power calculations.

6. PLOS authors have the option to publish the peer review history of their article (what does this mean? ). If published, this will include your full peer review and any attached files.

**Do you want your identity to be public for this peer review?** For information about this choice, including consent withdrawal, please see our Privacy Policy .

Reviewer #1: No

Reviewer #2: No

Reviewer #3: No

---

## [Decision Letter · Decision Letter 1]

15 Dec 2024

PGPH-D-24-01978R1

Postmortem point-of care hemoglobin testing is feasible and potentially accurate among children in South Africa

Dear Dr. Suchdev,

Thank you for submitting your manuscript to PLOS Global Public Health. After careful consideration, we feel that it has merit but does not fully meet PLOS Global Public Health’s publication criteria as it currently stands. Therefore, we invite you to submit a revised version of the manuscript that addresses the points raised during the review process.

The reviews are in general favor of the revised manuscript. However, Reviewer 3 identifies an important residual issue that should be addressed before your paper will be suitable for publication. I share Reviewer 3's assessment. Please consider this suggestion, and I look forward to receiving your revised manuscript.

We look forward to receiving your revised manuscript.

Kind regards,

Masako Fujita, Ph.D.

Academic Editor

Journal Requirements:

Additional Editor Comments (if provided):

Reviewers' comments:

Reviewer's Responses to Questions

**Comments to the Author**

1. If the authors have adequately addressed your comments raised in a previous round of review and you feel that this manuscript is now acceptable for publication, you may indicate that here to bypass the “Comments to the Author” section, enter your conflict of interest statement in the “Confidential to Editor” section, and submit your "Accept" recommendation.

Reviewer #1: All comments have been addressed

Reviewer #2: All comments have been addressed

Reviewer #3: (No Response)

2. Does this manuscript meet PLOS Global Public Health’s publication criteria ? Is the manuscript technically sound, and do the data support the conclusions? The manuscript must describe methodologically and ethically rigorous research with conclusions that are appropriately drawn based on the data presented.

Reviewer #1: Yes

Reviewer #2: Yes

Reviewer #3: Yes

3. Has the statistical analysis been performed appropriately and rigorously?

Reviewer #1: Yes

Reviewer #2: Yes

Reviewer #3: Yes

4. Have the authors made all data underlying the findings in their manuscript fully available (please refer to the Data Availability Statement at the start of the manuscript PDF file)?

Reviewer #1: Yes

Reviewer #2: Yes

Reviewer #3: Yes

5. Is the manuscript presented in an intelligible fashion and written in standard English?

Reviewer #1: Yes

Reviewer #2: Yes

Reviewer #3: Yes

6. Review Comments to the Author

Reviewer #1: Thank you for addressing my questions.

Reviewer #2: Thank you for your responses. I am satisfied that this paper is ready for publication.

Reviewer #3: This is an informative manuscript, suitable for publication. One this is still missing from their assessment, however. The authors need to provide a bit broader discussion to guide readers who might use their approach to identify anemia postmortem. They rightly caution these readers about the imperfect sensitivity and specificity of postmortem Hb for identifying peri-mortem anemia, but provide no cautions or guidance about attribution of cause and effect with regard to this peri-mortem anemia.

The study's goal is to advance understanding of how anemia contributes to child mortality. A huge part of this understanding must include assessing the role of anemia in disease progression. Not all children who die with anemia die because of anemia. Anemia can be a downstream consequence of many disease processes, in which case it is the disease and not the anemia that should be identified as the pressing public health target. (Managing disease-associated anemias may be an important goal for healthcare providers, of course, but this is distinct from public health and screening efforts.)

Mis-attributing deaths to anemia when children died with anemia, but because of a upstream disease, is thus a potential risk to use of their method that the authors of this paper should acknowledge in the Discussion. Even if they are not able to provide full and complete guidance for readers to distinguish between anemia that happened as part of disease progression and anemia that contributed to a death (a hugely complex and difficult assessment), they can caution readers that identifying anemia postpartum is only part of an assessment of whether anemia contributed to a death, ideally with helpful citations for such a thorough assessment.

7. PLOS authors have the option to publish the peer review history of their article (what does this mean? ). If published, this will include your full peer review and any attached files.

**Do you want your identity to be public for this peer review?** For information about this choice, including consent withdrawal, please see our Privacy Policy .

Reviewer #1: No

Reviewer #2: No

Reviewer #3: No

---

## [Decision Letter · Decision Letter 2]

6 Jan 2025

Postmortem point-of care hemoglobin testing is feasible and potentially accurate among children in South Africa

PGPH-D-24-01978R2

Dear Dr. Parminder Suchdev,

We are pleased to inform you that your manuscript 'Postmortem point-of care hemoglobin testing is feasible and potentially accurate among children in South Africa' has been provisionally accepted for publication in PLOS Global Public Health.

Best regards,

Masako Fujita, Ph.D.

Academic Editor

Reviewer Comments (if any, and for reference):

Reviewer's Responses to Questions

**Comments to the Author**

Reviewer #3: All comments have been addressed